# Using Open BIM and IFC to Enable a Comprehensive Consideration of Building Services within a Whole-Building LCA

**Sebastian Theißen [1,*], Jannick Höper [1], Jan Drzymalla [1], Reinhard Wimmer [2], Stanimira Markova [3], Anica Meins-Becker [4] and Michaela Lambertz [1]**

1   TH Köln (University of Applied Sciences), Institute of Building Services Engineering, Research Area Green Building, 50679 Cologne, Germany; jannick.hoeper@th-koeln.de (J.H.); jan_stefan.drzymalla@th-koeln.de (J.D.); michaela.lambertz@th-koeln.de (M.L.)
2   TMM Group Gesamtplanungs GmbH, Building Information Modeling R&D and Change Management, 71034 Böblingen, Germany; reinhard.wimmer@tmm-group.de
3   RWTH Aachen University, Institute for Building Design and Realization, Research area Future Technologies for Comprehensive Building Sustainability, 52062 Aachen, Germany; smarkova@bauplan.arch.rwth-aachen.de
4   University of Wuppertal, Interdisciplinary Centre III / Institute of Building Information Modeling, 42285 Wuppertal, Germany; a.meins-becker@uni-wuppertal.de
*   Correspondence: sebastian.theissen@th-koeln.de; Tel.: +49-221-8275-2871

**Abstract:** Holistic views of all environmental impacts for buildings such as Life Cycle Assessments (LCAs) are rarely performed. Building services are mostly included in this assessment only in a simplified way, which means that their embodied impacts are usually underestimated. Open Building Information Modeling (BIM) and Industry Foundation Classes (IFC) provide for significantly more efficient and comprehensive LCA performance. This study investigated how building services can be included in an open BIM-integrated whole-building LCA for the first time, identified challenges and showed six solution approaches. Based on the definition of 222 exchange requirements and their mapping with IFC, an example BIM model was modeled before the linking of 7312 BIM objects of building services with LCA data that were analyzed in an LCA tool. The results show that 94.5% of the BIM objects could only be linked by applying one of the six solution approaches. The main problems were due to: (1) modeling by a lack of standardization of attributes of BIM objects; (2) difficult machine readability of the building services LCA datasets as well as a general lack of these; and (3) non-standardized properties of building services and LCA specific dataset information in the IFC data format.

**Keywords:** open BIM; IFC; ER; LCA; LCA data; embodied impacts; building services; HVAC; MEP

## 1. Introduction

With the new EU Energy Performance of Buildings Directive (EPBD) 2018/844, the requirements for energy efficiency and the use of renewable energy sources in buildings are being tightened up at EU and national levels [1]. However, the problem with this development is that only the saving of energy consumption in the building operation phase is considered and environmental impacts resulting from production, dismantling and disposal are not taken into account. These embodied impacts become even more important when buildings are designed and built to a high energy efficiency standard [2]. Studies show that, for example, the share of embodied greenhouse gas (GHG) emissions can be responsible for up to 80% over the entire life cycle [3]. The relation between building construction and

building services is often not fully broken down, since the scope of a whole-building LCA (WBLCA) according to EN 15978 [4] is usually limited to the building construction and its main elements [5]. The main reason for this is that the effort required to determine the necessary data basis for an LCA of building services is very high [6]. The actual contribution of building services is therefore often unnoticed and underestimated [7–9]. The few studies in this research area show that, despite the low material mass share of building services of usually around 1–4% of the total building, it can be responsible for an unexpectedly high proportion of embodied impacts for new constructions of non-residential buildings [10]. In the case of GHG emissions, for example, $CO_2$ emissions are significantly higher than 20% or 30% [11–13]. Therefore, there is a high necessity to consider building services in the assessment of embodied impacts in a more integrated and coherent way with the building design [14].

There are limited possibilities to implement a holistic environmental assessment approach for buildings, and green building certifications are the only relevant option in terms of realistic and practical applicability. Green building certifications follow the WBLCA to evaluate the building environmental impacts in the context of its entire life cycle.

While the system of the German Sustainable Building Council (DGNB) or the Sustainable Building Assessment System (BNB) takes embodied impacts of building services into account within a WBLCA, other green building certification systems, such as the US-based LEED (Leadership in Energy and Environmental Design) [15] or the UK-based BREEAM (Building Research Establishment Environmental Assessment Method) International New Construction [16] do not consider embodied impacts of building services in their WBLCA requirements.

The BIM method in general and the Industry Foundation Classes (IFC) data exchange format used within the open BIM approach offer potential to perform WBLCAs significantly more efficiently and comprehensively [17]. By structuring the information required for calculations in a uniform way and making it more easily accessible within a digital building model, high automation potential for the implementation of a WBLCA is given, as shown by recent studies [18,19]. These potentials can be used especially for the simplification of data acquisitions of the building services within a WBLCA [20]. An essential part of this is the availability of LCA data. While most LCA databases provide LCA data for the architecture, there is a clear lack of LCA data for building services [12,21]. In addition, the technical setup/interface of the LCA database is important for linking information of the BIM model with environmental data [17].

The various approaches, in which BIM models are used to perform an LCA, generally export data from the BIM model for the Life Cycle Inventory (LCI) or perform a BIM integrated LCA based on a closed BIM approach [22]. This means that proprietary data formats of the respective software manufacturers are used, which produce isolated solutions that can be utilized only within the respective closed software environment. Additionally, in terms of LCA, these identified approaches only focus on architecture [23–28].

The reasons for this are, on the one hand, that there are large uncertainties in the early design phases of LCA, based on BIM models with low Level of Developments (LODs), and that minimal or no information content is available for building services [29,30]. On the other hand, the complexity and effort required for carrying out an LCA in late project phases, when an LOD of 300/400 is available, is very high [31]. Therefore, only a few studies have performed BIM based/integrated WBLCAs including a full consideration of building services [31].

There are even fewer solutions that are aimed at using the open BIM approach to perform a WBLCA while considering building services [32]. Besides research approaches, there are very few commercial tools that allow combining BIM and LCA by using an import of a BIM model as a native file of BIM software as well as an IFC or gbXML file. However, using the non-proprietary data format IFC has not been entirely developed yet for a fully automatic calculation of BIM-integrated WBLCA [26,32,33]. As a result, manual adjustments are required, making these approaches non-standard [17,34]. Considering the required effort through the complexity of conducting an LCI of building services within different

LODs of BIM models as well as the lack of LCA data of building services, it becomes even more difficult to perform an open BIM integrated WBLCA [17,26].

It is clear that there is a significant research gap addressing the full consideration of building services within an open-BIM and IFC-based whole-building LCA.

The overall aim of this paper is to contribute to a more intensive integration of building services in environmental assessments of buildings by using LCAs and the open BIM approach. This is particularly important as the need to expand the scope of WBLCA to include the current underestimated embodied impacts of building services is of great importance for assessing and optimizing buildings in a holistic view.

In this study, the linking of building services from heating, ventilation, air conditioning, refrigeration (HVACR) as well as plumbing, electrical and fire protection engineering with LCA data was implemented and tested for the first time within an open BIM approach using an example BIM model with LOD 300/400. The primary aim was to show how the calculation of embodied impacts of building services can be enabled based on an error free data exchange using IFC. Secondarily, it was investigated which different requirements between the linking of building services objects in BIM models and building services LCA datasets of ÖKOBAUDAT occur and how they can be harmonized to perform a correct and detailed WBLCA.

For demonstration purposes of the method, the following setup was chosen: Autodesk Revit in combination with liNear and other external BIM-object databases for the modeling; IFC as an open data exchange format; and ÖKOBAUDAT as an open source LCA database for building services and WBLCA requirements of green building certification systems. As a linking approach of LCA data, UUIDs of ÖKOBAUDAT LCA datasets were used to map them with the corresponding building services objects during the modeling process. An example BIM model was modeled according to VDI 6009, including 7312 BIM Objects of building services, to simulate an extensive and wide variety of building services systems to provide multiple requirements of the analyzation for linking these. Thereby, the different information levels of building services objects, which are output by the BIM software Autodesk Revit in combination with liNear and other external BIM-object databases, were analyzed and compared with the required information for linking building services LCA datasets of ÖKOBAUDAT. Semantic differences between generic and manufacturer specific BIM objects of building services were analyzed to enable functionally equivalent and automated links to LCA datasets of ÖKOBAUDAT.

As a result, missing or erroneous information from Autodesk Revit, ÖKOBAUDAT and incomplete data exchange processes using IFC were uncovered and solved, e.g., by data record supplements, such as extension of service lives or conversions to other reference units, in order to ensure a correct and automated linking of building services LCA datasets.

## 2. State of the Art

### 2.1. Standardization of Environmental Assessment According to EN 15978 and EN 15804

EN 15978 [4] and EN 15804 [35] are part of a series of standards that standardize the assessment of the overall performance of buildings throughout their whole life cycle. EN 15978 forms the basis for the assessment of the environmental performance using the LCA method. EN 15978 includes a description of the process for setting up the calculations required for the assessment. Once the goal and intended use of the evaluation have been determined, the object of the evaluation, the functional equivalent, the reference study period, the system boundaries and the building model with its physical properties are defined.

The norm also addresses the requirements on the structured collection of the quantity schedules and energy flow of every single building component as part of the comprehensive building quantification.

Besides the main elements of the building, technical systems and their services must also be included. An exemplary specification is provided for this purpose in supplementary Annex A of EN 15978.

However, the level of granularity chosen for the inclusion of physical properties ultimately depends on the individual objective and scope of the assessment and on the availability of data at the time of the assessment (sketch plan, design, procurement and handover) [4].

In terms to the objectives of this work, it is important to be able to examine the many individual requirements of building services components and their linking to LCA datasets. A very high level of detail is necessary for this. This is usually only available in late project phases or after completion, when the WBLCA and the documentation of the results are required as proof of certification. Therefore, the example BIM model (in Section 3.3) was created with a high level of detail.

To assess the environmental performance at the building level, information on building elements and construction products level is also required. Therefore, EN 15804 defines product category rules for the environmental product declarations of construction products.

In addition, EN 15804 standardizes which types of environmental impacts must be considered in which phases of a life cycle. Currently, 38 different (environmental) indicators are defined in DIN EN 15804:2020-03 that are to be considered in the life cycle phases: manufacturing phase (A1–A3), construction phase (A4–A5), service life phase (B1–B7) and disposal phase (C1–C4) [35]. The phases are supplemented by Module D, which includes information on reuse, recovery or recycling outside the life cycle and system boundaries (cradle to grave). A detailed overview is given in Figure 1.

| Life cycle stages | | | | | | | | | | | | | | | | |
|---|---|---|---|---|---|---|---|---|---|---|---|---|---|---|---|---|
| Product | | | Construction | | Use | | | | | | | End of Life | | | | Benefits |
| Raw material supply | Transport | Manufacturing | Transport | Construction installation process | Use | Maintenance | Repair | Replacement | Refurbishment | Operational energy use | Operational water use | Deconstruction demolition | Transport | Waste processing | Disposal | Reuse, Recovery, Recycling potential |
| Life cycle modules: A1 | A2 | A3 | A4 | A5 | B1 | B2 | B3 | B4 | B5 | B6 | B7 | C1 | C2 | C3 | C4 | D |
| Cradle to gate: x | x | x | | | | | | | | | | | | | | |
| Cradle to grave: x | x | x | x | x | x | x | x | x | x | x | x | x | x | x | x | (x) |

**Figure 1.** Life cycle stages and modules according to DIN EN 15978/15804.

While the version EN 15804:2014-07 requires only the declaration of Modules A1-A3 as obligatory for the manufacturers or the issuers of an Environmental Product Declaration (EPD), the declaration of all other environmental impacts in the remaining life cycle phases is optional [36]. As a result, only environmental impacts from the manufacturing phase are available in EPDs as LCA datasets in many cases. However, the new version DIN EN 15804:2020-03 requires that Modules A, C and D be considered as well.

## 2.2. LCA Data of ÖKOBAUDAT and IBU Data

In using the fully open approach to perform a comprehensive LCA in BIM here, only verified, open databased containing LCA relevant data were taken into consideration.

ÖKOBAUDAT is a database that uses LCA data to calculate WBLCAs. It was developed and is operated by the Federal Ministry of the Interior, Building and Homeland Affairs (BMI) and is widely considered as one of the largest verified and most reliable sources for building material related LCA data worldwide.

Another data source is IBU.data from the Institut Bauen und Umwelt (IBU), which is also openly available and free of charge [37].

ÖKOBAUDAT is the first standardized database that complies with DIN EN 15804 [38]. The data format of the ÖKOBAUDAT LCA datasets is based on the International Life Cycle Data System (ILCD) format with extensions for EPDs. It is a machine-readable exchange format for EPDs. For a unique identification, each dataset receives an identifier in the form of a Universally Unique Identifier (UUID,

a 16-digit hexadecimal number) as well as a version number. The version number is incremented when the dataset is changed [39]. The ÖKOBAUDAT is technically based on the software soda4LCA and is equipped with a standardized interface (API) for data exchange [40]. Via the interface, other applications and software tools can read data records from ÖKOBAUDAT or—with appropriate authorizations—import them directly into ÖKOBAUDAT [38].

In the current version of ÖKOBAUDAT (2020-II of 03.04.2020), 891 data records are available. The datasets are divided into four different dataset types:

- Generic dataset with a malus surcharge collected from different sources
- Representative—average dataset of selected manufacturers
- Average—average dataset of a manufacturer group
- Specific—manufacturer's data record

The ÖKOBAUDAT database consists to a large extent of generic datasets. About one third are EPDs in the current ÖKOBAUDAT. The IBU also provides more product-specific datasets for building products as standardized EPDs via IBU.data and consists of the same technical background as ÖKOBAUDAT.

In relation to building services, ÖKOBAUDAT provides 171 datasets. Furthermore, there are six EPDs which can be used for mapping building services.

### 2.3. Whole-Building LCA in Green Building Certifications Using DGNB System as an Example

The DGNB is considered the first system to have included WBLCAs within the environmental quality since 2007 [41]. Moreover, it also weights this equally with respect to the economic and sociocultural dimensions of sustainability within certification systems for buildings [42].

In the German DGNB system, the basis for WBLCAs is given by DIN EN 15978 and DIN EN 15804. In the criteria ENV1.1 "Building life cycle assessment", further requirements based on these standards are described to perform an WBLCA according to DGNB [43]. Besides the description of the goal, intended use and the functional equivalent, a service life of 50 years for the reference study period is also defined.

Furthermore, the system boundaries are set by considering the life cycle Modules A1-A3, B4, B6 and C3-C4 as well as Module D. As this paper focuses on embodied impacts, the operating energy in terms of Module B6 according to DIN EN 15804, e.g., in Germany via the Energy Saving Ordinance (EnEV), is not considered further here.

Within the DGNB system, a choice can be made between the simplified procedure (VeV) and the complete procedure (VoV) for the description of the physical characteristics of the building. While VoV basically stipulates a complete inclusion of nearly all components of the German cost groups (KG) 300 "building constructions" and KG 400 "building services" according to DIN 276 [44], VeV allows a limitation to eight essential component groups of building constructions and building services [7]. Specifically, VeV stipulates that the heating and cooling systems as well as the central units of the ventilation systems must be included in the building model. Pipes, ducts and other components of the HVACR system do not have to be included in the building model. As compensation for this simplification, the result of the environmental impacts in the individual life cycle phases must be "worsened" by multiplying a factor of 1.2, i.e., receive a 20% additional safety charge. If extensive passive measures are taken into account and recognized in the indicator passive systems of the criterion TEC 1.4 "Use and Integration of Building Services", the factor 1.2 can be reduced to 1.1 for passive buildings in the simplified procedure [43]. Since VoV requires a high manual effort, VeV is mainly used in the systems when calculating a WBLCA [7].

Within the Life Cycle Impact Assessment (LCIA), the following environmental indicators are required within the DGNB system: Global Warming Potential (GWP), Ozone Depletion Potential (ODP), Photochemical Ozone Creation Potential (POCP), Acidification Potential (AP), Eutrophication Potential (EP), Non-renewable primary energy demand (PEnr), Total primary energy demand (PEtot),

the proportion of renewable primary energy, Abiotic Resource consumption (ADP elements) and Fresh Water consumption (FW). The indicators ODP, ADPe and FW must only be presented in an informative manner. As data basis, the ÖKOBAUDAT version 2016-I or newer is prescribed.

For the service lives of the building structure, the service lives data spread sheet of the BNB System [45] is required to be used. The service lives of the building services are taken from VDI 2067 [46].

### 2.4. BIM Modeling

The BIM method integrates, links and enriches all relevant building data in a virtual data model for the entire life cycle of a building. As a holistic and cooperative working method, the BIM method enables not only 3D modeling but also an increasing depth of information and intelligence in digital building models [47]. Thus, BIM also offers a possibility to implement the information procurement and calculation process for the whole-building LCA in a structured and consistent manner to use automation potential based on this [18,48].

The BIM method requires a technological and cooperative architecture. Basically, BIM software is used for this. These comprehensive software tools enable an initial modeling of the 3D model and objects, enriched with data and information.

In addition, so-called information management or model checker tools are used, which allow an extended processing of the data model. Furthermore, these tools allow referencing or rather linking different BIM models of different subjects by following the method of multimodeling [49].

Multimodeling is a method for linking different models of various subject areas as one unit [50]. Multimodels consist of heterogeneous models and explicit, external links between their elements. This means that the connection between the different models is managed and controlled by so-called link models. To provide a comprehensive basis for the WBLCA, the linking of architectural and building services model is therefore of particular importance and can be realized with the help of the multi model method [17].

For this purpose, data must be exchanged continuously during the individual life cycle phases of a building, e.g., between project participants in an adequate form. If these data are incomplete or contains errors, this will lead to insufficient data exchanges and cause problems during further use. Therefore, the structured exchange of data from the building data model, in addition to the model-based graphical representation of the building, is a central part of the BIM methodology. Thus, BIM offers the possibility of structuring and exchanging the data captured in the model in open and non-proprietary data formats, such as the Industry Foundation Classes (IFC) developed by the buildingSMART organization, standardized as DIN EN ISO 16739 [51].

### 2.5. Open BIM and Industry Foundation Classes (IFC)

The scheme of the IFC data format is based on the data modeling language EXPRESS (ISO 10303-11), which uses the language standard for the exchange of product model data (STEP) for file-based exchange. In addition to this data format, an exchange can also take place via ifcXML, ifcZIP, Construction Operations Building Information Exchange (COBie), gbXML or LandXML files. The XML data exchange format (eXtensible Markup Language) primarily maps weather data, usage profiles, geometric and energy-relevant data, whereby parameters relevant to building technology are sometimes neglected. The IFC data format is currently a widely used basis for the open BIM approach to enable data exchange between different software applications as a uniform, manufacturer-neutral data format [52]. The object-oriented data format developed by buildingSMART contains a hierarchy, whereby information can be inherited from top to bottom. Elements of a higher level can therefore refer to elements of a lower level, but not vice versa. In its architecture, the IFC data format is fundamentally divided into four levels:

- Domain layer (department level): The domain layer contains specialized classes that can only be located in this domain (e.g., HVACR domain). In the hierarchy structure, this layer is located at the

highest level, which means that these classes cannot be referenced by other layers. For example, the domain-specific classes for the building services or architecture are located in this layer.

- Interop layer (interoperability layer): This layer often contains classes used by different actors involved in the construction process. Fundamental classes, such as IfcWall or IfcRoof, are defined here.
- Core layer: The core layer contains the most significant classes in the IFC data model. Furthermore, the root class is defined here by the kernel schema. This schema also contains definitions for relationships, products, processes, etc.
- Resource layer: The classes integrated in the lowest level do not have their own Globally Unique Identifier, so these classes can only be referenced and cannot reference other classes. The classes contain basic information, such as quantities or geometry.

The IFC format is further divided into classes and types. Classes group together objects with the same or similar properties and types can specify these properties. For example, a volume flow controller is assigned to the IFC class "IfcAirTerminalBox". However, as there are different types of these, a detailed semantic description can be provided by defining the type, e.g., constant flow volume flow controller (IfcAirTerminalBoxTypeEnum.CONSTANTFLOW). Currently, 776 classes and 2027 types are available in DIN EN ISO 16739, of which 62 classes can be used for building services components [53]. The IFC data format is strictly separated between semantic description and geometric representation.

The definition of objects, relationships and properties is the focus of the IFC format. Furthermore, a distinction is made between real and abstract objects. A real object is, e.g., a wall or table, i.e., objects from the real world. Abstract objects are, e.g., processes or roles in a project. In the IFC data model, these definitions are generally concretized by so-called entities (Entity), functions (Functions), rules (Rules), attributes (Attributes) and relations (Relations). In addition, there are the functions of quantity lists (QuantitySets) and dynamic properties (PropertySet).

In the current version of the IFC data format (IFC4 Add2), the Psets PSet_Environmental ImpactIndicators [54] and PSet_EnvironmentalImpactValues [55] can be used to integrate environmental impact information.

In addition, extendable property sets can be defined modularly. Some of these property sets are already listed in the schema documentation of the data model. In this way, specification-independent property sets that go beyond the schema definition can be assigned to the individual classes, allowing individualized semantic enhancement functions [56]. Data types and value ranges from the state characteristics of the classes and can be further specified by enumerations and relations to other objects.

*2.6. LCA-BIM Integration Strategies with Focus on Building Services*

In general, there are two major strategies for connecting LCA and BIM: A BOQ export (Bill Of Quantity) based on the BIM model, for import in native LCA-software; and a real time LCA-calculation within a closed or open BIM approach, giving real time feedback of the calculated results [57].

Since the first main strategy creates a static process that only works in one direction, it causes a separation between model and analysis, which subsequently leads to non-dynamic or non-interoperable systems [58].

There are many different approaches to enable real-time evaluations within the second main strategy. One of the most important aspects is how data can be integrated into the model. For example, all information can be integrated into the model as an integral part [27].

Another possibility is, if the technical requirements are available, e.g., API and record history of an LCA database, to establish links to external LCA databases, which can be managed even if the native BIM model is changed [17]. Such an approach was pursued in the research project: "Life Cycle Assessment and BIM in sustainable construction".

In another work based on this approach, Höper developed a method and a tool that can perform dynamic WBLCAs, especially in late project phases [53].

The IFC data format was used for model exchange in combination with user defined property sets (UDPS) from Autodesk Revit. The option of Autodesk Revit's UDPS enabled the precise definition and transfer of the required information. This method allows the transfer of information that is not standardized in IFC, as well as standardized properties.

Furthermore, these alphanumeric data are also needed for the functionally equivalent and correct linking of building services datasets of ÖKOBAUDAT. For example, the specification of the performance of a ventilation system is important in order to enable the correct link with an ÖKOBAUDAT dataset. A linkage with purely geometrical information is therefore not sufficient in many cases. However, the required semantic information is usually only known with the design building services systems. In case of ventilation systems, it is only possible to determine the dimensions (height and width) of the ventilation ducts after the design. With this information and the information about the material, mostly galvanized steel sheet, the weight in kilograms per meter can be calculated and a functionally equivalent link to the ÖKOBAUDAT dataset can be established.

The few available software tools, which enable building services modeling, e.g., Autodesk Revit, do not allow detailed design and calculation of building services. Therefore, the calculation associated with building services design requires interfaces to specific TGA software, such as liNear, DDS-CAD, or Solar Computer [53].

## 3. Method and Materials

As mentioned above, the main aim of this research was to investigate how the linking of data with the ongoing calculation of embodied impacts of building services can be enabled within an open BIM approach based on an error free data exchange using IFC.

The procedure is divided into nine steps as shown in Figure 2. First, the building model is defined, based on EN 15978 and DGNB requirements, for a WBLCA, including a comprehensive consideration of the building services. Based on this, information exchange requirements (ERs) are defined in Step 2. These are mapped with the current standardized parameters in the IFC data model.

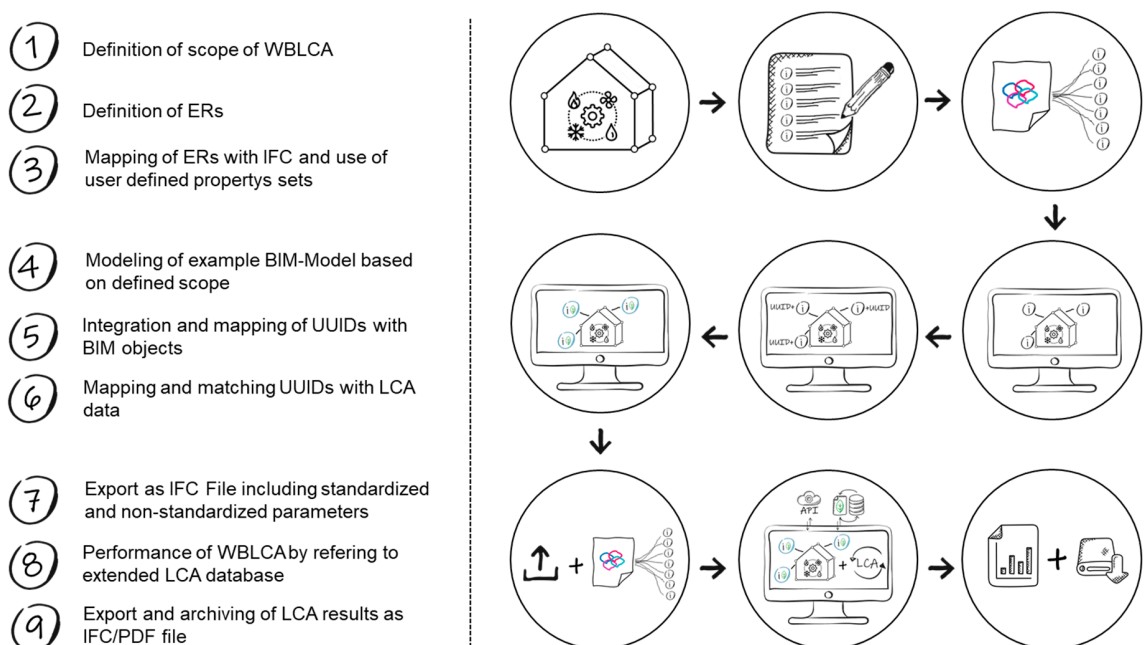

**Figure 2.** Schematic diagram of integrating and linking building services within an open BIM integrated whole-building LCA.

In doing so, 222 ERs according to ISO 29481 were defined and mapped to the IFC data model in Step 3. If new, non-standard characteristics or attributes are defined, they must be manually added to the BIM model as project parameters.

In Step 4, an example BIM model was created that contains all building services components as BIM objects according to the defined scope of Step 1.

The architecture and building services of the example BIM model were modeled in LOD 300/400 with Autodesk Revit according to the VDI example building of VDI 6009 [59]. Besides the library of Autodesk Revit, the libraries of liNear and BIM-objects from external databases were used. During the modeling process, an exact dimensioning of the building services system was neglected, as only the correct linking of the ÖKOBAUDAT datasets was important to achieve the defined objective, rather than the assessment of the absolute results of the embodied impacts.

During modeling, the respective BIM objects of building services were extended with the parameter "UUID" in Step 5 and manually assigned to the BIM objects in the following Step 6.

For further use of the BIM model, an IFC export is necessary in Step 7. This IFC export is performed using the integrated export function within the BIM software. In addition, the export of the UDPS must be selected to enable the data exchange of non-standard ERs.

As soon as the export is available as an IFC file that contains all necessary data, the WBLCA is performed in Step 8. To enable this, the IFC file is imported into the LCA software. In this case, DESITE BIM and the LCA tool developed by Höper based on DESITE BIM were used for this purpose.

The data basis of this tool is ÖKOBAUDAT and extracts of the IBU.data as well as service life data of VDI standard 2067 for building services [46].

Since an explicit assignment of the BIM objects to a ÖKOBAUDAT LCA dataset UUID is available from Step 6, the UUIDs of the objects can be compared with the integrated database via rule-based linking. If the UUID of the building services components shows a match, a link is established, and data can be accessed for further calculations. In this way, the object receives the environmental information and can be used for the WBLCA. At this point, the application programming interface (API) interface of ÖKOBAUDAT or IBU.data could be used, too. However, this procedure was not applied due to missing information within the datasets, e.g., service lives.

In Step 9, the results are provided as an IFC file or PDF report.

### 3.1. Information Delivery Manual/Exchange Requirements/Model View Definition

An Information Delivery Manual (IDM) is an ISO standard used by the International buildingSMART Organization that formalizes the information requirements in the collaboration process between the different project participants for a specific purpose.

The IDM serves as a basis for a standardized definition of processes, identifying interfaces between the relevant project roles and deriving Exchange Requirements (ERs), which serve as the basis for the Model View Definition (MVD).

For a unified understanding, the IDM and MVD methodology is standardized in DIN EN ISO 29481-1 and -2 [60,61]. In general, the five "W questions" should be answered in an IDM: Who needs what information and in what data format from whom when? This method allows a quick overview of the process and the associated sub-processes for a specific use case through the diagram. Thus, specialized data and information are defined, standardized and presented only for the use case.

The ERs are derived from the IDM for each interface identified in the process diagram. After the process and the associated information supply chains are defined, the required information to be exchanged can be specified. The ER can also be understood as data requirement specifications, which serve as a basis for the LODs. This involves a detailed specification of information which role has to provide the ER. Furthermore, the mapping of the building information for the used data model, e.g., IFC, can also be carried out specifically for each interface in its according ER.

Finally, the MVD can be developed from the modules mentioned. The MVD method was developed to filter the mass of information of a digital building model and to display only the relevant information for a specific BIM use case.

Therefore, this harmonized IDM and MVD methodology allows a component-based formalization of a BIM use case from the management level to software implementation [62].

For example, for the WBLCA use case, not all of the information that is enriched in the building process is relevant. An MVD therefore specifies the subset of the model that is relevant for the respective use case.

### 3.2. Implementation Use Case—Autodesk Revit/liNear/BIM-Object Databases/DESITE BIM

Autodesk Revit is an object-oriented planning tool for architects, structural engineers and building technicians that supports BIM technology. Autodesk Revit enables both 2D and 3D modeling of a component-oriented building model and enables data exchange via the industry standard formats RVT, DWG, DXF and DWF. In addition, it offers open BIM data exchange using data formats such as IFC. The Autodesk Revit interface for export and import of IFC files is officially certified for all standardized, published by buildingSMART MVDs.

In addition to selecting the MVD when exporting the BIM model as an IFC file, UDPS can be defined using a text file. This option allows exporting specific attributes that are not defined in IFC.

Autodesk Revit also allows the implementation of plug-ins that enable various data enrichments and communication with other specialized program add-ons, such as liNear.

LiNear from the company liNear GmbH is a planning software for building services and is available as a plugin, e.g., for Autodesk Revit. liNear enables the calculation of heating or cooling loads as well as pipe and duct systems calculations. In addition, liNear has its own object library of building services components that can be used for BIM modeling.

BIM-objects are described by a geometric representation and a semantic information content and can be addressed by a unique Global Unique Identifier (GUID) in BIM models. Since many objects can occur in BIM models, they are described more specifically by the three levels of class, type and specimen. BIM-objects are assigned in BIM libraries, which are available in BIM modeling software or in external databases, e.g., "bimobject" or "plan.one". They usually have direct interfaces to the BIM modeling programs available on the market and the IFC data format.

DESITE BIM from thinkproject GmbH is an information management program [63]. In contrast to Autodesk Revit, this tool cannot be used for geometrical modeling. Instead, information is processed. This software allows a holistic access to all data and information in the model and enables the individual evaluation of data through its open interface architecture.

In doing so, DESITE BIM supports the model checking process and provides access to the information of the model. Thus, models imported in the IFC format, for example, can be checked, analyzed and extended in many ways based on self-programmed checking rules.

With the help of so-called modules and domains, the data are accessed and individual information is extracted from the model and further processed. Thus, external data based on the XML/CSV format can be directly integrated and processed with the help of DESITE BIM. Furthermore, external databases such as ÖKOBAUDAT and building services relevant EPDs from the IBU.data can be integrated with DESITE BIM.

The tool allows the import of different models of different subjects that can be combined to a coordination model. DESITE BIM has an API that can be used to create new object properties, automated program sequences and WebForms. Especially via the WebForms, a "program" can be developed in the actual program. The programming interface can be addressed and edited using JavaScript.

*3.3. Modeling of an Example BIM model as a Semantically Rich Reference for Building Services*

The following section describes the case study and the assumptions made. Since no real projects as BIM models with the needed variety of building services were available as real cases, an example BIM model was created as shown in Figure 3.

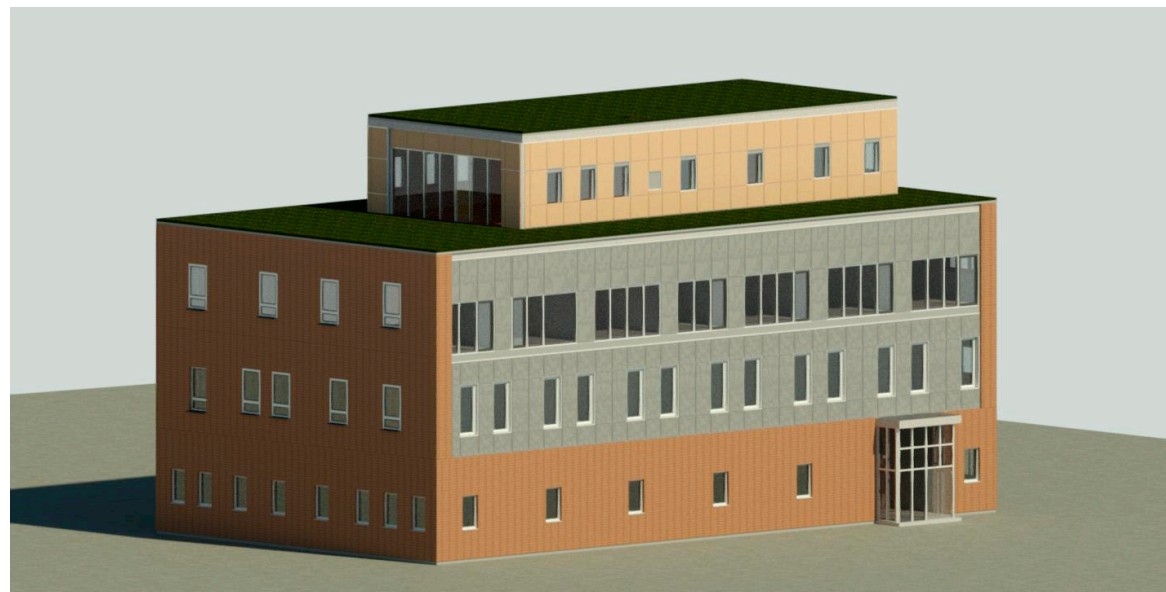

**Figure 3.** The architecture of the example BIM model in Autodesk Revit.

The architecture and the building services objects of the office building were modeled according to the VDI 6009 [59]. It was modeled with Autodesk Revit in combination with liNear and external BIM-object databases. It has a level of development (LOD) above 300 [53]. The example BIM model consists of four floors and an underground parking level. Total gross floor area is 2228 m$^2$, whereby the net floor area is 1849 m$^2$. The building structure can be described as: a reinforced concrete supporting structure with basement, external walls made of reinforced concrete with core insulation and clinker brick façade as well as curtain walls made of aluminum (mullion and transom façade). The windows were modeled in different variants (aluminum, plastic and wood) and triple glazed. The reinforced concrete flat roof with sealing is partially greened. Materials and components of the architecture were roughly specified and superficially modeled, as the focus of the example BIM model was primarily on the correct linking of a wide variety of building services components in BIM models with ÖKOBAUDAT LCA datasets. Therefore, it was the aim to represent all reference units of the ÖKOBAUDAT datasets of building services as requirement profiles to be examined.

The primary objective was therefore to represent as many technical components of building services equipment as possible from heating, ventilation, air conditioning and refrigeration (HVACR) as well as plumbing, electrical and fire protection engineering to simulate a large number of different building service components in the BIM model. Building automation was not possible to represent as there were no LCA datasets within ÖKOBAUDAT.

For this reason, the modeling of the BIM model also neglected the exact dimensioning of the system technology. Thus, different types of installation and supply were modeled for the different floors. While, for example, the heat transfer is transmitted by different types of radiators on the ground floor and the first floor, the other floors supply heat by underfloor heating.

In addition, different materials for pipes and ducts were chosen to investigate possible differences in linking with LCA datasets of ÖKOBAUDAT. A similar procedure was used for the ventilation system: rectangular and round ducts were considered. External insulation was considered for all pipes and ducts.

Refrigeration was produced by the use of a chiller and was transferred to the room via cooling ceilings. The refrigeration distribution network was modeled as a Tichelmann system [64]. For the sanitary system, in addition to the pipes for fresh water and wastewater, the fittings were also taken into account. Furthermore, lamps, sockets, switches, cables and a photovoltaic system on the façade were modeled. Finally, smoke control dampers, a passenger lift, circulation pumps and volume flow controllers were integrated.

A representation of the modeled building services is shown in Figure 4. A summary of the important parameters of the building services is described in Table 1.

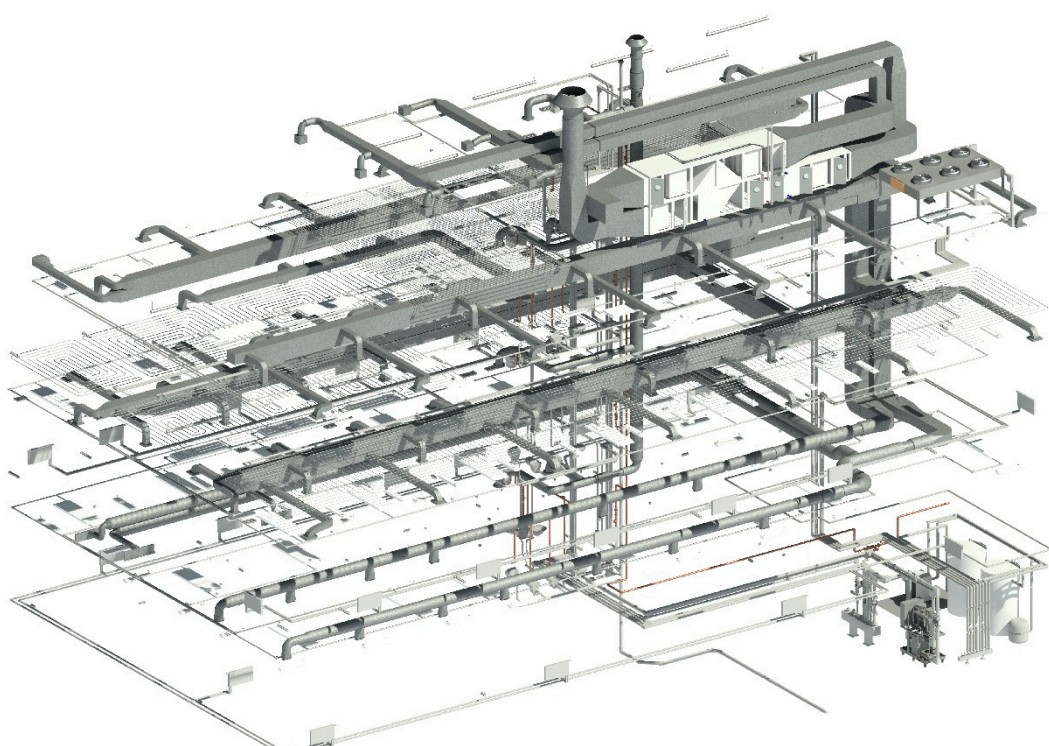

**Figure 4.** The building services systems of the example BIM model in Autodesk Revit.

**Table 1.** Description of the building services.

| **Hot Water Preparation: Decentralized Hot Water Preparation Using Electric Continuous Flow Heaters** |
|---|
| Heat generator: Gas condensing boiler<br>Heat distribution network: hot water storage tanks; heating manifold; circulating pumps; different sizes (DN), materials (PE-X, stainless steel, copper) and insulation material (EPDM foam) of pipes; expansion tank<br>Space heating first floor: different types of radiators; installation of distribution network was modeled as a Tichelmann system<br>Space heating second floor: different sizes of radiators; installation of distribution network was modeled as standard installation (same distance of supply and return flow)<br>Space heating other floors: different types of underfloor heating depending on material (PEX, copper) and installation distance (100 mm, 200 mm) |
| Sanitary installation: different sizes (DN), material (copper) and insulation material (rockwool) of pipes for fresh water; different sizes (DN) of pipes for wastewater, material of pipes for wastewater (Acrylonitrile butadiene styrene, ABS); sanitary ceramic (toilets and fittings) |
| Ventilation: central ventilation system with heat recovery (cross-flow heat exchanger)<br>Ventilation distribution network: air ducts (zinc coated steel plate) with different sizes (DN), shapes (round, rectangular) and insulation material (rockwool); damper (flow); smoke control damper; sound absorbers, filters, volume flow controllers, flexible air duct (PVC)<br>Ventilation supply: ceiling diffusers |
| Refrigeration: refrigeration plant (underground level); chiller (located on roof)<br>Refrigeration distribution network: different sizes (DN); material (steel) and insulation material (flexible elastomer); cooling manifold; circulation pumps; installation of distribution network was modeled as a Tichelmann system<br>Refrigeration supply: cooling ceilings |
| Independent power supply installations: photovoltaic system consisting of PV module, cabling, fasteners and inverter<br>Low voltage installation equipment: sockets; switches; cables<br>Lighting systems: multiple spots; pendants; reflectors<br>Passenger transport: one lift |

In terms of LCA, DGNB requirements were applied. Besides a reference study period of 50 years, system boundaries of Modules A1-A3, B4, C3-C4 and D based on DIN EN 15804 were considered. Module B4 only includes the creation and disposal of the replaced product, not the replacement process itself [43]. The replacement cycles for building services were taken from VDI 2067. Since this study only focused on embodied impacts, Module B6 was not a point of interest.

### 3.4. Dynamic LCA User Tool and Database Structure within DESITE BIM

Based on the results of the research project "Life Cycle Assessment and BIM in sustainable construction" [65], a dynamic LCA user tool and database was developed within DESITE BIM [53].

The tool was developed based on the workflow presented in Section 3. This means it sets specific modeling requirements, e.g., the integration of UUIDs, and requires the export of the BIM model as an IFC file including UDPS.

The user tool consists of a front-end user interface that was developed using JavaScript, JQuery and HTML. The calculation algorithms were written as scripts in JavaScript. Furthermore, a database was integrated into DESITE as a CSV file. The data basis is based on ÖKOBAUDAT (Version 2020-I), building services relevant EPDs from IBU.data and additional developed extensions.

At this point, other LCA databases can also be used: The Swiss KBOB LCA database or the Austrian baubook Eco2soft database also provide free accessible building services LCA data for different environmental impacts indicators. The UK based Inventory of Carbon & Energy (ICE) Database and the Australian-based EPiC database are also open source databases for building materials, but are limited to embodied energy, carbon and water and do not provide specific LCA datasets for building services. Similarly, the Quartz Common Product Database also offers open source LCA data for around 100 building materials, but not for building services. In contrast to the open source databases, GaBi and Ecoinvent databases contain over 12,000 ready-to-use Life Cycle Inventory profiles. Here, limitations are given due to their fee-based availability.

In the field of building services, for example, the service lives of VDI 2067 were added. In total, the integrated database contains 1456 datasets.

As a result, a real-time calculation of environmental impacts and variants at building, component and product level is possible.

The user interface shows the environmental impacts of the indicators GWP, AP, EP, POCP, PEnr and PEtot given in their respective unit per year and area. It was extended with regard to further analysis options to include the output of information on the IFC class, component ID, layer number, material type from the BIM model, area, layer thickness, linked dataset and service life.

Editing or manual adjustment of materials, service lives, component surfaces and layer thicknesses are also possible. By selecting objects, individual components of the building or groups can be visually highlighted and displayed with direct calculation of the environmental impacts as shown in Figure 5.

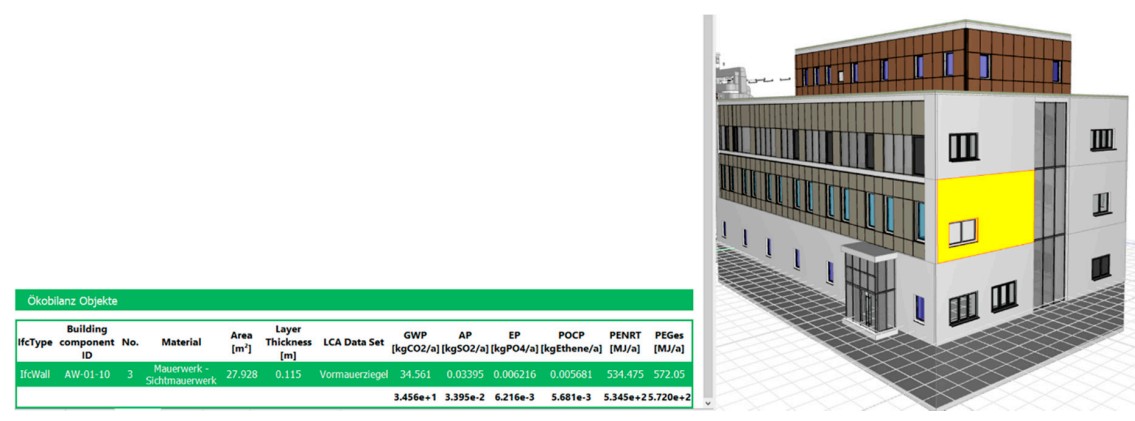

**Figure 5.** User interface in DESITE BIM, which shows the selection of an external wall.

Specific filtering of IFC classes is possible via an additional function in the user interface. For example, it is possible to filter according to IFC classes of building services to select pipes, ducts or radiators and calculate their environmental impacts.

In Figure 6, the left side of the screenshot shows the user interface of the tool in DESITE BIM that displays the environmental impacts of radiators.

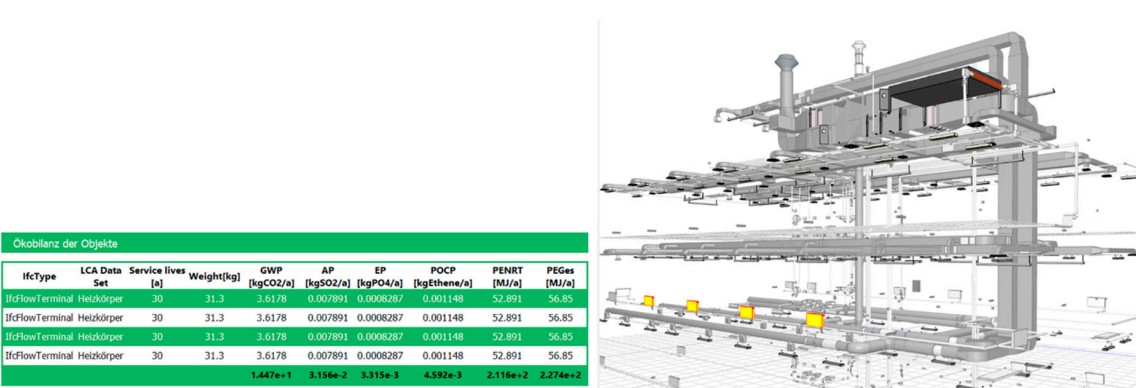

**Figure 6.** User interface in DESITE BIM, which shows the selection of the radiators.

## 4. Results

By performing the described method for achieving the objectives, results were achieved in four areas: Definition of the ERs for the comprehensive consideration of building services, ensuring the comprehensive export of them, solution approaches for correct linking and finally the absolute number of linked building services LCA datasets.

### 4.1. Definition of Exchange Requirements

To investigate how building services can be fully considered within a WBLCA and which challenges have to be solved in the context of the linkage, ERs according to the set scope have to be defined. For this purpose, ERs of building services were first identified and specified.

The framework defined in DIN EN 15978 served as a template for the description of the physical building model. The building services systems were classified according to DIN 276.

To provide a sufficient basis for the development of an MVD by defining and classifying the content of the building services, further technical information types needed to be defined for data exchange via IFC [62]. In addition to the type of subject of the model, specification of the component/element or unit, information on the respective PSet, property or Qset is also important. In total, 19 different types of information were used to define and map the ERs with the IFC data model.

In this respect, the extent to which the required information is already standardized in the IFC data model was checked. For each attribute of an element, a check was first made to see whether there was a suitable property set, quantity set or concept for it. If that was true, the corresponding IFC class was assigned to the attribute, followed by the property set and finally the property with the corresponding type and unit. For attributes that represent masses or quantities, such as kilograms, the quantity sets of IFC were used. The equivalent quantity description (Quantity) and type (QuantityType) were assigned to the attribute.

Finally, the materiality was represented by the concepts. The Material Single concept was used for building services objects, such as pipe bends, that consist of only one material. The Material Layer Set concept was used for objects, for example, hot water storage tank, that are composed of several materials.

Furthermore, when specifying the ERs for building services, the reference units of the ÖKOBAUDAT datasets, mostly with reference to kilograms, were considered. This procedure was chosen to simplify the link between the datasets and BIM objects.

As a result, 222 ERs were defined and mapped with the IFC data model. Overall, 209 of the 222 ERs could be assigned to standardized properties. The missing attributes were described by self-developed property and quantity sets: Plca_ and Qlca_. Figure 7 shows an extract of the defined ERs. The complete list is attached to the publication as Supplementary Materials—S1.

| Process responsibility | Category | Element (according to DIN 276) | Attribut | Unit | Export (M mandatory, O optional, - no export) | Entity | Concept | Geometry | PSet |
|---|---|---|---|---|---|---|---|---|---|
| MEP-Engineer | Building services systems | Sewerage systems | material of sewage pipe | - | O | IfcPipeSegment | Material Layer Set | | |
| MEP-Engineer | Building services systems | Sewerage systems | mass of sewage pipe | kg | O | | | | |
| MEP-Engineer | Building services systems | Sewerage systems | length of sewage pipe | m | O | | | | |
| MEP-Engineer | Building services systems | Sewerage systems | nominal diameter of the sewage pipe | DN | O | IfcPipeSegment | | | Pset_PipeSegmentTypeCommon |
| MEP-Engineer | Building services systems | Sewerage systems | material of the sewage pipe insulation | - | O | IfcCovering | Material Layer Set | | Pset_CoveringCommon |
| MEP-Engineer | Building services systems | Sewerage systems | volume of the sewage pipe insulation | m³ | O | | | Volume Quantity | |
| MEP-Engineer | Building services systems | Sewerage systems | material valve | - | O | IfcValve | MaterialSingle | | |

**Figure 7.** Selection of defined ERs of building services within a WBLCA mapped to IFC 2x3.

### 4.2. Enabling IFC-Export According to Defined ERs Using User Defined Property Sets

Due to the lack of standardization within the IFC data format for building services and other environmental information needed for the LCA data linking of ÖKOBAUDAT, UDPS had to be defined for Autodesk Revit, as otherwise the data exchange within the open BIM approach would be insufficient and a WBLCA with a comprehensive consideration of building services could not be performed [53].

The reason for this is the availability of existing MVDs during data export. Since the current MVDs transfer only a part of the required data in the sense of this work, important data would be lost during the export. To avoid this loss of data, the IFC export must be extended.

For this purpose, BIM software, in this case Autodesk Revit, offers the possibility to correctly locate information per UDPS, self-defined Psets, such as the Plca_Lca including the UUID, to attributes using a mapping table in the IFC schema. In doing so, attributes were manually assigned to the IFC classes (e.g., IfcDuctSegment) in the UDPS. The table must be available in the form of a text file (*.txt) and can be used during IFC export [66]. During the IFC export, all properties defined within this mapping table are transferred and can be interpreted by other IFC-conform programs. When declaring the UDPS, defined, formal principles for data transfer must be observed.

This process simulates the use of an implemented LCA MVD for Autodesk Revit in a simplified way. Thus, the open BIM approach can be maintained, but information that is not standardized according to buildingSMART is also transferred.

### 4.3. Solution Approaches to Link BIM Objects of Building Services with LCA Datasets of ÖKOBAUDAT

After a complete, error-free IFC export of the example BIM model with all defined ERs could be realized, the different problems concerning the correct linkage of building services datasets of ÖKOBAUDAT were solved by different solution approaches with regard to the primary goal.

These applied approaches can be divided into the three areas as shown in Table 2: BIM software, ÖKOBAUDAT and IFC data model. Six approaches were used, which are briefly described below.

**Table 2.** Overview of applied solution approaches to enable correct linking with ÖKOBAUDAT LCA datasets.

| | Solution Approach | Description |
|---|---|---|
| | **BIM Software** | |
| 1 | Recalculation of hollow cylinders | Autodesk Revit uses a formula to calculate the volume of hollow cylinders that is based on the volume of a full cylinder instead of considering it as a hollow cylinder. This results in a deviation of approximately 44% from the correct calculation for the volume of a hollow cylinder (see Appendix A). The same was also found out for the insulation of cuboids (angular ventilation ducts). As a result, too much mass is calculated for pipe and duct insulation. To solve this problem, a database of duct and pipe dimensions was developed. Depending on the dimensions, the height and the width or the diameter were integrated. |
| 2 | Harmonization of machine readability by standardizing of attributes | A key issue of linking manufacturer-specific BIM objects is the focus on the geometric representation of the manufacturer, which means that the objects cannot be divided into different development levels. Furthermore, the semantic description of objects is currently not standardized. Thus, they are provided in a different scope and structure depending on the software/library used. Even within Autodesk Revit, parameters are named differently. For example, the angle specification for bends for pipes is given as "angle" and for ventilation ducts as "alpha". For these reasons, a classification for the UDPS was designed to allow machine readability. |
| 3 | Creation of an inheritance logic of pipes and pipe insulation | To harmonize the service lives of connected building services components, a database was created. For example, if a pipe has a standardized service life of 30 years and the pipe insulation is expected to have a service life of 40 years, the service life of the insulation must be set to 30 years. For this purpose, a parameter was defined within Revit, since the UUID of the pipe cannot be inherited by the associated insulation. Therefore, a parameter was developed to simulate the inheritance. Further, a database with the corresponding information was integrated into DESITE BIM. Finally, an algorithm was developed to dynamically define the service life of the pipe. |
| | **ÖKOBAUDAT** | |
| 4 | Completion of service life data | Datasets of ÖKOBAUDAT provide only in very few cases information about the service lives, e.g., VDI 2067. Therefore, all building services ÖKOBAUDAT datasets were assigned the corresponding services lives of VDI 2067 with the help of the integrated database structure of DESITE BIM. |
| 5 | Adaptation of machine readability by extension of the database | In addition to the environmental information, basic data on the mapping of building services were also enriched. Within ÖKOBDAUDAT, the building services components are primarily related to the mass, but since this information is mainly missing in BIM models, databases were developed for this purpose. Furthermore, the weight of, e.g., pipes, varies depending on DN and material. Within the corresponding dataset, weight tables are stored as figures as background information. Thus, although the necessary information for linking is available for a dataset, machine readability for different design formats, defined as different DNs for pipes and ventilation ducts, is significantly more difficult. Therefore, the weight tables were used to develop a database extension. Instead of one dataset, it was split up into various datasets for each DN and material. Similar to this, other databases were also developed, e.g., for radiators depending on their size. In total, eight databases were integrated within DESITE BIM. |
| | **IFC** | |
| 6 | Use of user defined property sets | The current IFC structure and MVDs do not allow the integration of ÖKOBAUDAT specific types of information, such as UUIDs or the descriptions of background systems. Therefore, the IFC export of Autodesk Revit was extended using UDPS. In doing so, an extended MVD was simulated, e.g., to perform the data exchange of UUIDs for explicit linking. |

*4.4. Total Amount of Linked LCA Datasets with Building Services BIM Objects*

Using the model building, 6884 BIM objects out of a total of 7312 objects could be linked. The breakdown in Table 3 shows which BIM objects were used after application of the six solution approaches in order to correctly link the BIM objects with the building services ÖKOBAUDAT dataset.

As a result, a total of 29 different datasets of building services could be linked by applying the developed solution approaches. In total, 428 objects could not be linked because the needed 12 datasets were not available in ÖKOBAUDAT. Cables could not be linked due to the missing geometrical representation in the BIM model, although there is an ÖKOBAUDAT dataset.

The building automation could not be considered in general, because there are no datasets available within ÖKOBAUDAT.

**Table 3.** Results of linked LCA datasets with building services BIM objects.

| LCA Dataset of ÖKOBAUDAT | Linkable without any Solution Approach | Linkable with Use of Solution Approach | Not Linkable with Use of Solution Approach | Total Number of Objects | Solution Approach Applied | | | | | |
|---|---|---|---|---|---|---|---|---|---|---|
| | | | | | 1 | 2 | 3 | 4 | 5 | 6 |
| Air filters | 0 | 0 | 2 | 2 | | | | | | |
| Air ventilation duct (zinc coated steel plate) | 0 | 482 | 0 | 482 | | x | | x | x | x |
| Blank copper drinking pipe | 0 | 359 | 0 | 359 | | x | | x | x | x |
| Cables | 0 | 0 | 0 | 0 | | | | | | |
| Ceiling diffusers | 0 | 0 | 111 | 111 | | | | | | |
| Chiller | 0 | 0 | 1 | 1 | | | | | | |
| Circulating pump < 50 W | 5 | 0 | 0 | 5 | | | | x | | x |
| Cooling ceilings | 0 | 0 | 54 | 54 | | | | | | |
| Damper (flow) | 0 | 0 | 87 | 87 | | | | | | |
| Drinking water pipe (stainless steel) | 0 | 321 | 0 | 321 | | x | | x | x | x |
| Drinking water pipe PE-X | 0 | 1014 | 0 | 1014 | | x | | x | x | x |
| EK90 smoke control damper (EK92 series) | 28 | 0 | 0 | 28 | | | | x | | x |
| Electric continuous flow heater (21 kW) | 7 | 0 | 0 | 7 | | | | x | | x |
| Electric socket | 113 | 0 | 0 | 113 | | | | x | | x |
| Elevator basic component (independent of floor) | 1 | 0 | 0 | 1 | | | | x | | x |
| EPDM foam (pipe insulation); EPDM foam | 0 | 1372 | 0 | 1372 | x | | x | x | | x |
| Expansion tank | 0 | 0 | 1 | 1 | | | | | | |
| Flexible air duct (PVC) | 0 | 0 | 61 | 61 | | | | | | |
| Gas low temperature boiler 20–120 kW (upright unit) | 1 | 0 | 0 | 1 | | | | x | | x |
| Hot water storage tank (stainless steel) | 0 | 2 | 0 | 2 | | | | x | x | x |
| LED office luminaire | 17 | 0 | 0 | 17 | | | | x | | x |
| Louvre light integrated into ceiling 2x T5-28W (LFL) | 78 | 0 | 0 | 78 | | | | x | | x |
| Louvre light integrated into ceiling 3x T5-14W (LFL) | 54 | 0 | 0 | 54 | | | | x | | x |
| Louvre light integrated into ceiling 3x T8-18W (LFL) | 34 | 0 | 0 | 34 | | | | x | | x |
| Manifold heating and cooling | 0 | 0 | 2 | 2 | | | | | | |
| Mineral wool pipe jacket; Rock wool | 0 | 326 | 0 | 326 | x | | x | x | | x |
| Photovoltaic system 1000 kWh/m$^2$ *a | 0 | 20 | 0 | 20 | | | | x | | x |
| Radiator; Painted flat radiator of steel | 0 | 25 | 0 | 25 | | x | | x | x | x |
| Refrigeration plant | 0 | 0 | 1 | 1 | | | | | | |
| Rocker light switch | 52 | 0 | 0 | 52 | | | | x | | x |
| Rockwool | 0 | 703 | 0 | 703 | x | | x | x | | x |
| Sanitary ceramic | 0 | 21 | 0 | 21 | | x | | x | x | x |
| Sanitary fittings | 0 | 0 | 87 | 87 | | | | | | |
| Sewer pipe (Acrylonitrile butadiene styrene - ABS) | 0 | 208 | 0 | 208 | | x | | x | x | x |
| SH/Armaflex insulation for industrial and building installation | 0 | 797 | 0 | 797 | x | | x | x | | x |
| Sound absorbers | 0 | 0 | 21 | 21 | | | | | | |
| Steel pipe | 0 | 800 | 0 | 800 | | x | | x | x | x |
| Underfloor heating system copper (100 mm distance) | 0 | 10 | 0 | 10 | | x | | x | x | x |
| Underfloor heating system PEX (200 mm distance) | 0 | 20 | 0 | 20 | | x | | x | x | x |
| Ventilation centralized with heat recovery per 10,000 m$^3$/h | 1 | 0 | 0 | 1 | | | | x | | x |
| Volume flow controller VRE, VR | 13 | 0 | 0 | 13 | | | | x | | x |
| **Total** | **404** | **6480** | **428** | **7312** | | | | | | |

## 5. Discussion

In the following, the results listed above are discussed and classified for further research and standardization processes.

### 5.1. Use of ERs for Further IDM Development and MVD Implementation

From a technical point of view, the ERs describe the information that has to be exchanged between two processes of different actors at a certain point in time in the use case when information is transferred. The ERs must be defined in a way that a technical implementation in BIM software systems and exchange formats, to be generated by these, can be ensured [67]. Therefore, these 222 identified ERs

were "mapped" with the IFC data model. The extent to which this information can already be mapped or described by the IFC data model was investigated. The mapping process of the ERs with the IFC data model thus represents the location of the identified information for the WBLCA with a comprehensive consideration of building services in the IFC structure.

Since 13 of the 222 identified ERs could not be assigned in a standardized way in the IFC data model, UDPS were developed. As IFC is based on a standardized language, only the requirements for the declaration of information have to be met. From a technical point of view, all attributes of a digital building models can therefore be transferred using the IFC data format. However, the definition of the attributes must always be coordinated, e.g., within a project consortium.

Therefore, the non-standardized ERs but can be seen as a start for extending the IFC data model to define missing important properties, especially for building services, in the future. In Germany, for example, this standardization is managed by VDI 2552 Part 11.4 [68] or the buildingSMART BIM and Sustainability Group [69] and is managed internationally by the buildingSMART Use Case Management [70].

Furthermore, the ERs could be used in further research or standardization processes to develop IDMs and, based on this, MVDs. Since ERs only form one part of an IDM, it is still important, for example, to define the tasks and dates for the purpose of information enrichment. This is necessary to implement the technical verifiability of the minimum data exchange and levels of detail in BIM modeling programs.

In terms of an MVD for the use case presented, only the necessary information for the WBLCA including the building services components are filtered and exchanged. Thus, the ERs that were defined in this work form an important basis for including building services components within an open BIM integrated WBLCA and their use in MVDs as control instruments during modeling.

To this end, with the further development and use of IDMs in the future, there will be a rising demand of their use in BIM software. As a result, it is very likely that software manufacturers will be faced with the task of implementing the results of these processes as MVDs in their BIM software systems. With regard to the objective, the defined ERs will contribute to the standardization of buildings services relevant for WBLCA in the IFC data format.

### 5.2. Technical Characteristics Using Autodesk Revit and ÖKOBAUDAT LCA Datasets

While modeling of architecture requires certain modeling guidelines, e.g., monolithic vs. layered modeling of a wall [17], there are also some technical characteristics that have to be considered when modeling building services within Autodesk Revit.

One of them is for example electrical cables. Due to the missing geometric representation within the BIM model, these could not be evaluated. Another important aspect is the assignment of building services to IFC classes. It should be noted that the logic of Autodesk Revit combines different building services objects into "HVAC components", e.g., heat generators, HVAC units or radiators and exports them as IfcBuildngElementProxy to the IFC file. This generalization means that IFC classes cannot transfer specific information. This results in data loss, which is important for the performance of a WBLCA.

Furthermore, it became clear during the building services modeling with Autodesk Revit that the volume of insulation in hollow cylinder bodies is not correctly determined [53]. As a result, the mass determination is incorrect and the calculation of environmental impacts of the insulation is not calculated correctly. This is especially important for solutions that use a BOQ export based on Autodesk Revit for the comprehensive LCA of building services engineering.

Difficulties can also arise during data transfer when using external BIM-object databases. These objects are declared by specific manufacturers, which makes it difficult to standardize the parameters. Moreover, these BIM-objects also cannot guarantee the correct class assignment in the sense of IFC. Thus, manufacturer objects are mainly transferred as "IfcBuildngElementProxy". As a

consequence, the geometric representation is present in the model, but an explicit assignment is not made. The required information for a WBLCA of the respective object is not transferred.

Another aspect of using BIM-objects is the non-standardized information content of objects. This allows manufacturers to decide what information the object should contain. The creation of a separate database in this work was therefore important to be able to standardize and add missing information.

In the future, a standardized information content is also considered to be useful here in order to avoid individual solutions and to promote practical applications. For building services, the VDI 3805 series of guidelines for "Electronic Product Data Interchange in the building services " provides an ideal basis for orientation [71].

Furthermore, in terms of LCA data, an extension of ÖKOBAUDAT in the field of building services is considered necessary in the future as certain BIM Objects could not be linked because of the lack of datasets. The following datasets are proposed for the inclusion of new datasets that are currently missing or that did not appear relevant following simplified calculation methods for WBLCA, e.g., DGNB:

- Air diffusers
- Air filters
- Chiller
- Cooling ceilings/heating ceilings
- Damper (flow)
- Expansion tank Sensors/actuators of building automation
- Flexible air duct
- Fuel cell
- Ice storage
- Lithium-ion power battery
- Manifold heating and cooling
- PCM elements
- PV modules separated in monocrystalline and polycrystalline, inverter
- Refrigerating plants (absorption, compression, etc.)
- Sanitary fittings
- Ventilation system components (sound absorbers, etc.)

In addition, EPDs as Armaflex or CLIMAFLEX pipe insulation could be included in order to provide a broader or more product-specific data basis.

If a more comprehensive LCA database of building services were available, it would also be possible to determine the environmental impact of the building services more comprehensively. This would result in an improved environmental assessment of building services in general.

Furthermore, it can be summarized that the better databases for LCA data become technically accessible, the easier it will be to map and link datasets with BIM objects within the open BIM approach and other external databases in the construction sector. While from a technical point of view, many structures already exist, questions remain open, for example with regard to responsibilities in the area of mapping and maintenance. The standardization process, e.g., ISO 22057, will also be important in this respect, as it generally deals with how EPD data can be made more useful for environmental assessment with BIM.

*5.3. Potential of Proposed Method*

In general, this developed method with (semi)automated linkage and calculation significantly facilitates a high level, comprehensive LCA of building services, which has been currently neglected due to a lack of data, excessive manual effort and an often underestimated influence on the environmental performance of a building.

The principle of integrating UUIDs of datasets during modeling requires manual effort and specific expertise in ÖKOBAUDAT, BIM modeling and the IFC data model. These requirements and previous workflows do not form a practical basis and are rather a starting point for further future research and standardization processes.

The implementation within the open BIM approach showed many difficulties and, in a strict sense, could only be implemented to a limited extent as an "open" solution approach.

To close the challenge of modeling and gaps in datasets or the IFC data model, many adjustments and extensions were necessary, which were implemented with the help of six solution approaches and an own database. In addition, the effort for the WBLCA could be reduced, e.g., by avoiding having to manually integrate weight data of pipes into the model.

Furthermore, linking at the material level requires that the materiality must be clearly defined. Thus, this method tends to offer a helpful tool only in late project phases and should therefore be seen more as a simplification of the WBLCA as a certification proof, e.g., in the DGNB system, in order to fully calculate WBLCA with full consideration of building services. While there is a very high effort for this [6], e.g., by considering the distribution networks of pipes and ducts with all components, the developed method can simplify the linking of ÖKOBAUDAT datasets with the high number of pipe and duct system parts.

Nonetheless, this method cannot be directly applied for an early phase optimization of the environmental impacts. However, the basic principle of linking datasets via a unique assignment (UUID/GUID) and access using an API interface is still considered to be useful.

It can be applied the same way to a "building component mapping" in early planning phases as the detailed "material mapping" in late project phases. LCA benchmarks could be used as preliminary estimates [72,73].

In addition to the way of integrating a UUID during modeling as presented here, UUIDs could also be directly contained in databases that contain preconfigured BIM-objects. Basically, the ÖKOBAUDAT datasets in ILCD data format already provide a good starting point.

## 6. Conclusions and Outlook

The authors believe that building services need to be more addressed in environmental assessments of buildings when using the whole-building LCA, as they contribute to a meaningful share of embodied impacts, especially when designing and constructing high energy efficient buildings. However, as WBLCAs with detailed consideration of building services are very rarely performed, due to their high complexity and extensive effort required, this necessity is still neglected and/or not known.

With a reduced complexity and effort for conducting comprehensive WBLCAs including building services, the authors see a high potential and motivation for considering building services. Here, open BIM and open data exchange formats form the basis for such progress.

Therefore, the main objective of this study was to show which issues have to be faced and solved in order to enable a comprehensive consideration of building services within a whole-building LCA using open BIM and IFC. This study focused for the first time on the full environmental assessment of building services within an open BIM integrated WBLCA and its technical challenges. A method was developed that first defined the information exchange requirements for a comprehensive consideration of building services within a whole-building LCA and their mapping with an open data exchange format. Secondly, an example BIM model was modeled, based on the defined scope, to be able to map and link building services objects with LCA data. Thirdly, an export of the BIM model was generated using an open data exchange format, to be able to perform the whole-building LCA with full consideration of building services in an LCA tool.

The results show that only 5.5% of the 7312 BIM objects could be linked directly without any adaptive actions. This was because the belonging LCA datasets provided the functional unit "pieces". By developing and applying at least one of the six solution approaches, 94.5% of the 7312 BIM objects of building services could be correctly linked with LCA datasets. Here, mainly the functional

units "kilogram" or "cubic meter" of the LCA datasets caused problems, e.g., the conversions to the corresponding unit coming out of the BIM model. The remaining 5.5% could not be assigned mainly due to the lack of LCA data for specific building services. In addition, notable particularities are the missing geometrical representation of electric cables in the BIM model created with Autodesk Revit as well as the missing LCA data for building automation.

The six solution approaches still require high manual effort, which is not practicable in a real design process. Therefore, the results should be seen more as starting points that point out where adaptations are needed and useful: BIM Software (Revit), open exchange data format (IFC) or LCA database (ÖKOBAUDAT). The developed ERs can also be used for the developed IDMs and MVDs or be used for the mapping process of other open data exchange formats.

In addition, to be able to calculate environmental impacts of building services within an open BIM integrated WBLCA, the approach of using UUIDs of LCA datasets and their link to BIM objects is seen as an advantage with further research potential, as performance of BIM models can be kept very high and responsibilities of data quality and availability is managed by the database provider.

The authors see a strong need for further research, especially in the question of how LCA data, especially of building services, should be provided and integrated within BIM models and which ERs should be required depending on the different LODs to enable open BIM integrated WBLCAs during the design process.

**Supplementary Materials:** The following are available online at http://www.mdpi.com/2071-1050/12/14/5644/s1, File S1: Definition of ERs of building services within WBLCA mapped to IFC 2x3.

**Author Contributions:** Conceptualization, all authors; methodology, J.H., S.T., R.W., and S.M.; software, J.H.; formal analysis, J.H., S.T., and J.D.; investigation, S.T. and J.H.; writing—original draft preparation, S.T. and J.H.; writing—review and editing, all authors; visualization, S.T.; and supervision, M.L., A.M.-B., R.W., and S.M. All authors have read and agreed to the published version of the manuscript.

**Funding:** This research received no external funding.

**Conflicts of Interest:** The authors declare no conflict of interest.

## Appendix A. Recalculation of Hollow Cylinders

**Table A1.** Recalculation of hollow cylinders.

| | | |
|---|---|---|
| **Initial parameters** | length = 1 m | (1) |
| | Da pipe = 0.022 m | (2) |
| | di pipe = 0.02 m | (3) |
| | D insulation = 0.02 m | (4) |
| **Calculation hollow cylinders, REVIT** | = (Da pipe + (2 * D insulation)) * $\pi$ * length * D insulation | (5) |
| | = (0.022 m + (2 * 0.02 m)) * $\pi$ * 1 m * 0.02 m | (6) |
| | = 0.0039 m$^3$ | (7) |
| **Calculation hollow cylinders, real** | = ((Da pipe + (2 * D insulation))$^2$ − di pipe$^2$) * $\pi$/4 * length | (8) |
| | = ((0.022 m + (2 * 0.02 m))$^2$ − 0.02 m$^2$) * $\pi$/4 * 1 m | (9) |
| | = 0.0027 m$^3$ | (10) |

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
