# Peer review of "Using Open BIM and IFC to Enable a Comprehensive Consideration of Building Services within a Whole-Building LCA"

_sustainability, doi:10.3390/su12145644_

Round 1

Reviewer 1 Report

The paper is discussing the importance of BIM which is a cutting-edge software of the modern construction industry for the consideration of building services life cycle. I recommend minor revisions.

Abstract:

In general, abstract is written well but it can be further enhanced by adding novelty of this study. I mean how this is a new kind of study or how it is different from the previous studies related to the same research problem. Additionally, for the statement “The results show that various adaptations of BIM software, LCA data, and IFC were required to enable the comprehensive consideration of building services within the open BIM WBLCA”, not sure what is adaptations, you can mention the most important adaptation here.

Introduction:

“However, using non-proprietary data 89 formats such as IFC, have not been entirely developed yet for a fully automatic calculation of BIM-90 integrated WBLCA, for instance [26,32,33].” ---- As per one recent article it may be possible, you can refer it “Conceptualizing “COBieEvaluator”: A rule-based system for tracking asset changes using COBie datasheets.

Material and methods:

Text of Image 5 and 6 is not visible.

Conclusion:

This section is not written well. What are the main findings of your research? You need to summarize the content and purpose of your research paper. “With an ongoing technological development combined with a tightening of the energy efficiency standard of buildings, a higher significance of embodied impacts in the field of building services can be expected in the future” ---- Is this an outcome of your research?

Reviewer 2 Report

Thank you for submitting your paper “Using open BIM and IFC to enable a comprehensive consideration of building services within a whole building LCA” to the Journal of Sustainability.

The study is very interesting and overall it is well written but needs some improvements.

  1. I suggest reorganizing the abstract, it should report the main numerical results. The originality of the paper needs to be stated clearly showing the most significant results.
  2. In the introduction, it is necessary to put in evidence the real innovation of the present study. The novelty of the study is not very clear. The approach of the study is not very well explained in the introduction.
  3. In general, the contribution of each mentioned article should be defined, especially for articles grouped together, where the single contribution is unclear.
  4. Table 1 can be summarized reporting the principal technical specifications of the systems used and the thermal parameters of the envelope, the description can be reported in the main text.
  5. In general, it is important to put in evidence the following points:

- The proposed methodology has numerous advantages, such as:

- The main novelties and objectives of the present work can be summarized as:

  1. Figures 5 and 6 are not easily readable
  2. Table 2 should be summarized.

Reviewer 3 Report

The study is well structured, precise and well referenced.

However, I don't want readers to perceive any limitations in the study; limits given by the lack of a real case study, by the application of too specific dates (LCA data of the German official LCA database ÖKOBAUDAT).

For this reason, I would suggest, in chapter 3, to find the space for a brief comparison with other LCA databases.

I would ask to improve the graphic quality of figures 4 and 6 for the 3d model.

I would add some figures with a plan and a section with dimensions to better understand the case study.
